# Downregulation of SOX9 expression in developing entheses adjacent to intramembranous bone

**Asahi Kitamura[1], Masahito Yamamoto[2,3]\*, Hidetomo Hirouchi[3], Genji Watanabe[3], Shuichiro Taniguchi[3], Sayo Sekiya[3], Satoshi Ishizuka[4], Juhee Jeong[5], Kazunari Higa[6], Shuichiro Yamashita[1], Shinichi Abe[3]**

1 Department of Removable Partial Prosthodontics, Tokyo Dental College, Chiyoda-ku, Tokyo, Japan, 2 Division of Basic Medical Science, Department of Anatomy, Tokai University School of Medicine, Kanagawa, Japan, 3 Department of Anatomy, Tokyo Dental College, Chiyoda-ku, Tokyo, Japan, 4 Department of Pharmacology, Tokyo Dental College, Chiyoda-ku, Tokyo, Japan, 5 Department of Basic Science and Craniofacial Biology, New York University College of Dentistry, New York, NY, United States of America, 6 Ophthalmology/Cornea Center, Tokyo Dental College Ichikawa General Hospital, Ichikawa, Chiba, Japan

\* masahitoyamamoto0412@gmail.com

**Data Availability Statement:** All relevant data are within the manuscript and its Supporting Information files.

## Abstract

Entheses are classified into three types: fibrocartilaginous, fibrous, and periosteal insertions. However, the mechanism behind the development of fibrous entheses and periosteal insertions remains unclear. Since both entheses are part of the temporomandibular joint (TMJ), this study analyzes the TMJ entheses. Here, we show that SOX9 expression is negatively regulated during TMJ enthesis development, unlike fibrocartilage entheses which are modularly formed by SCX and SOX9 positive progenitors. The TMJ entheses was adjacent to the intramembranous bone rather than cartilage. SOX9 expression was diminished during TMJ enthesis development. To clarify the functional role of *Sox9* in the development of TMJ entheses, we examined these structures in TMJ using *Wnt1Cre;Sox9^{flox/+}* reporter mice. *Wnt1Cre;Sox9^{flox/+}* mice showed enthesial deformation at the TMJ. Next, we also observed a diminished SOX9 expression area at the enthesis in contact with the clavicle's membranous bone portion, similar to the TMJ entheses. Together, these findings reveal that the timing of SOX9 expression varies with the ossification development mode.

## Introduction

The enthesis is where tendons and ligaments meet bone, serving as a critical junction for movement and load transmission [1, 2]. They typically categorized into three types: fibrocartilaginous, fibrous, and periosteal insertions [3]. The dense fibrous connective tissue, uncalcified fibrocartilage, calcified fibrocartilage, and bone comprise the four transition zones anchoring the fibrocartilaginous enthesis to the bone. In fibrous entheses, the tendon or ligament directly attaches to the bone through dense fibrous connective tissue. In periosteal insertions, muscle fiber bundles attach directly to the periosteum without tendons. Several groups demonstrated histological and molecular mechanisms of mature entheses [4–7].

**Funding:** This study was supported by a grant for Assistance in Joint Research (20K10191: Masahito Yamamoto and 20K09895: Shinichi Abe) with the Community Program in Life Sciences from the Ministry of Education, Culture, Sports, Science and Technology of Japan Masahito Yamamoto: Conceptualization, Preparation of the manuscript, Data curation, Formal analysis, Investigation, Resources, Shinichi Abe: Project administration, Resources, Decision to publish.

**Competing interests:** The authors declare no competing or financial interests.

Because no specific marker of tendon development has yet been identified, it has long been unclear how tendon-bone interfaces develop. In the early 2000s, several groups demonstrated that scleraxis (SCX), a bHLH transcription factor, is expressed throughout tendon differentiation [8, 9]. To determine the function of Scx, Murchison et al. [10] generated Scx$^{-/-}$ mice and found that they exhibited severe tendon defects. This led to further investigations of the role of Scx in development of the tendon-bone interface. Blitz et al. [11] showed that bone morphogenetic protein 4 (BMP4) acts downstream of Scx, and that deletion of Bmp4 in tendons has a marked effect on enthesis development. In 2013, two research groups demonstrated that progenitors double-positive for Scx and Sox9 (a regulator of cartilage formation) form entheses under the regulation of TGFβ and BMP4 signaling [12, 13]. In mice, Schwartz et al. [14] revealed a population of Hedgehog-responsive cells in the developing enthesis that is distinct from tendon and chondrocytes. Although previous studies have clarified the development of fibrocartilaginous entheses, little is known about that in "fibrous entheses" and "periosteal insertions".

The temporomandibular joint (TMJ) is a hinge-type synovial joint that connects the mandible (lower jaw) to the temporal bone. Specifically, it forms an articulation between the condylar process (mandible) and the mandibular fossa (temporal bone). The articular disc is a thin oval plate composed of non-vascular fibrous connective tissue located between the two [15, 16]. The lateral pterygoid muscle, one of the four masticatory muscles, attaches to the anterior part of the condylar process and the articular disc [17, 18]. Entheses in the human TMJ are unique because they comprise all types of entheses [19]. However, it is unclear how they develop in the TMJ. This study aimed to clarify the development of "fibrous entheses" and "periosteal insertions" by investigating the TMJ.

## Materials and methods

### Experimental animals

All mouse experiments were conducted following the National Institutes of Health guidelines for animal care and use. The experiments received approval from the Tokyo Dental College Institutional Animal Care and Use Committee (protocol #240106). We used C57BL6J mice at embryonic days (E) 13.5, 14.5, 15.5, 16.5, 17.5, 18.5, and at 12 weeks old. *Wnt1-lacZ* (B6CBA-Tg(Wnt1-lacZ)206Amc/J), *Wnt1$^{Cre}$* (129S4. Cg-E2f1Tg(Wnt1-cre)2Sor/J), *Sox9$^{flox/flox}$* (B6.129S7-Sox9tm2Crm/J), *Sox9$^{creER}$* (STOCK Tg(Sox9cre/ERT2)1Msan/J), and *R26$^{tdTomato}$* (B6;129S6-Gt(ROSA)26Sortm14(CAG-tdTomato)Hze/J) mice were obtained from the Jackson Laboratory (Bar Harbor, ME, USA). All mice were raised in specific pathogen-free conditions. *Sox9$^{creER}$* mice were mated with *R26$^{tdTomato}$* mice to produce *Sox9$^{creER}$; R26$^{tdTomato}$* mice. *Wnt1$^{Cre}$* transgenic mice were mated with *Sox9$^{flox/flox}$* mice to generate *Wnt1$^{Cre}$; Sox9$^{flox/+}$* mice. We performed PCR to genotype each strain following the Jackson Laboratory guidelines. A female mouse was housed with a male overnight, and noon on the day a vaginal plug was noted was designated as E0.5. For the harvesting of embryos, timed-pregnant females were sacrificed by CO2 intoxication. The gravid uterus was dissected out and suspended in a bath of cold phosphate-buffered saline, and the embryos were harvested after amnionectomy and removal of placenta. The adult mice were also euthanized by using CO2 intoxication.

To trace the lineages of cells expressing *Sox9* during the development of enthesis and to generate time-specific Sox9 knockout mice, we generated double-transgenic *Sox9creERT2/R26tdTomato* mice and *Wnt1CreER/Sox9floxed/+* in which Cre expression in *Sox9$^{creERT2}$/R26$^{tdTomato}$* mice and *Wnt1$^{CreER}$/Sox9$^{floxed/+}$* in which Cre expression in *Sox9$^+$* and *Wnt1$^+$* progenitor cells could be induced at different developmental stages by administration of tamoxifen (Tam) [20].

## Double staining of Alkaline Phosphatase (ALP) and desmin

Sections were stained using an ALP staining kit (Primary Cell, Hokkaido, Japan) following the manufacturer's instructions. In summary, the sections were rinsed in running distilled water for 1 minute, after which 50 mL of staining solution was applied to each section. The sections were incubated at room temperature for 3 hours until the ALP turned dark blue, then washed in PBS. Next, the slides were incubated in methanol with 3% hydrogen peroxide for 30 minutes. The sections underwent multiple additional washes in PBS and were then incubated with 3% bovine serum albumin for 1 hour to block non-specific binding. Next, sections were treated with primary desmin antibody (dilution, 1:1000; Abcam) and incubated overnight at 37˚C in a humid chamber. The secondary antibody was applied using the EnVision™+ Dual Link System-HRP (Dako, Tokyo, Japan) at room temperature. Next, following multiple PBS washes, the sections underwent impact DAB staining (Funakoshi) for reaction detection and were then examined post hematoxylin counterstaining. This staining method is deemed promising for further studies and clarification of musculoskeletal system development [20].

## Jaw movement function test

Physiological experiments on jaw movements were conducted right after removing female mice at E14, E16, and E18. For embryo collection, pregnant females were euthanized with $CO_2$ inhalation. We identified the frequency of jaw movements in one minute. At each stage, a video was recorded using a smartphone (iPhone 13 mini, Apple Inc., California, USA).

## RNA in situ hybridization

The antisense probe for *Scx* has been previously described [21]. The probe was labeled with digoxigenin (DIG RNA labeling mix, Roche, Rotkreuz, Switzerland), and hybridization was carried out following a standard protocol [22]. In brief, sections were fixed for 10 minutes with 4% paraformaldehyde, digested with 1 Ug/ml proteinase K (Roche) for 5 minutes, and then refixed for 5 minutes. Acetylation was carried out for 10 minutes using a solution of triethanolamine, hydrochloric acid, and acetic anhydride. The sections were pre-blocked with hybridization buffer (50% formamide, 5x SSC, 50 mg/ml yeast tRNA, 1% SDS, 50 mg/ml heparin) and then incubated with the antisense probe*Scleraxis (Scx)* at 1 ng/ml dilution in the hybridization buffer. After washing the unbound probes with SSC buffer, the probes on the sections were detected using anti-digoxigenin antibody conjugated to alkaline phosphatase (Roche) and developed with BM purple (Roche). The images were analyzed using Image-Pro (Media Cybernetics, MD, USA).

## Tamoxifen treatment

We dissolved tamoxifen (T5648; Sigma-Aldrich, St. Louis, MO, USA) in ethanol and then diluted it in corn oil (C8267; Sigma-Aldrich) to a 10 mg/mL concentration, as previously described. Next, we administered 1.5 mg or 3 mg of tamoxifen into the peritoneal cavity of pregnant mice at E13 or E15, respectively, and co-injected 1 mg/40 g of progesterone (P8783; Sigma-Aldrich).

## Immunohistochemical analysis

To assess protein expression in the samples, immunolocalization of type II collagen (COLII), SRY-box 9 (SOX9), runt-related transcription factor 2 (RUNX2), and desmin was examined. The COLII, SOX9, and RUNX2 slides were incubated and digested with 25 mg/ml testicular hyaluronidase (Sigma Chemicals, St Louis, MO, USA) in PBS for 1 hour at 37˚C. After

multiple PBS washes, the sections were incubated in methanol with 3% hydrogen peroxide for 30 minutes, followed by additional PBS washes and a 1-hour incubation in 3% bovine serum albumin to block non-specific binding. Subsequently, sections were incubated with primary antibodies for COLII (1/400, LSL, Tokyo, Japan), desmin (1/1000, Abcam, Cambridge, UK), RUNX2 (1/1000, Santa Cruz, Texas, USA), and SOX9 (1/1000, Abcam) for 24 hours at 4°C in a humid chamber (S1 Table). Next, the secondary antibody was applied using an ABC staining kit (Funakoshi, Tokyo, Japan) with the following secondary antibodies: mouse anti-mouse immunoglobulin G (IgG) Alexa Fluor 488 (dilution 1:1000; Thermo Fisher Scientific, Waltham, MA, USA) and goat anti-mouse IgG Alexa Fluor 555 (dilution 1:1000; Thermo Fisher Scientific) (S2 Table). We treated some sections with ImmPACT 3,3′-diaminobenzidine (DAB) (Funakoshi) to reveal any reactions, and then examined the sections post hematoxylin counterstaining. The images were analyzed using Image-Pro (Media Cybernetics, MD, USA).

### BrdU labeling/histology

A 10 mg/ml BrdU stock (Abcam) was administered intraperitoneally to mice on embryonic days (E) 14, 15, and 16 at a dose of 100 μg BrdU per gram of body weight. The mice were sacrificed one day post injection, and the embryos were harvested. BrdU staining was performed on paraffin sections using a BrdU staining kit following the manufacturer's protocol (Zymed, South San Francisco, CA).

### Reverse transcription-polymerase chain reaction

We collected the muscle-bone attachment site (including the enthesis) (Fig 3M) and the condyle alone (Fig 3N) for RNA purification. RNA was extracted using the SV Total RNA Isolation System (Promega, WI, USA) and cDNA was synthesized using avian myeloblastosis virus reverse transcriptase (Takara Bio Inc., Shiga, Japan). To investigate the mRNA expression of the Sox9 and Runx2 genes related to entheses, reverse transcription-polymerase chain reaction (RT-PCR) analysis was conducted. Glyceraldehyde-3-phosphate dehydrogenase (GAPDH) was used as an internal standard. RT-PCR cycling conditions included 30 cycles consisting of thermal denaturation at 95°C for 30 s, annealing at 52°C for 30 s, and extension at 72°C for 20 s, followed by a final extension at 72°C for 5 min. Gel electrophoresis was carried out at 100 V for 20 min. Primer sequences and product sizes for the RT-PCR and primer assay ID and amplicon length for real-time PCR are shown in S3 Table.

### Statistical analysis

All statistical analyses were conducted using EZR (Saitama Medical Center, Jichi Medical University, Saitama, Japan), a GUI for R (The R Foundation for Statistical Computing, Vienna, Austria). More precisely, it is a modified version of R Commander designed to incorporate frequently used statistical functions in biostatistics. The data were deemed statistically significant with $P < 0.05$.

## Results

### Mouse's TMJ lacks fibrocartilage between the LPM and pterygoid fovea

Before exploring the TMJ enthesis development, we examined the adult enthesis in C57BL6J mice. The mouse's condyle had a mushroom-like structure (Fig 1A–1D), with a depression in the medial aspect of the condylar neck (Fig 1D), the pterygoid fovea, connected to the lateral pterygoid muscle (LPM) (Fig 1D and 1E). The intramuscular and extramuscular tendons extended from the pterygoid fovea (Fig 1E–1G). The first was part of the LPM's core (Fig 1G),

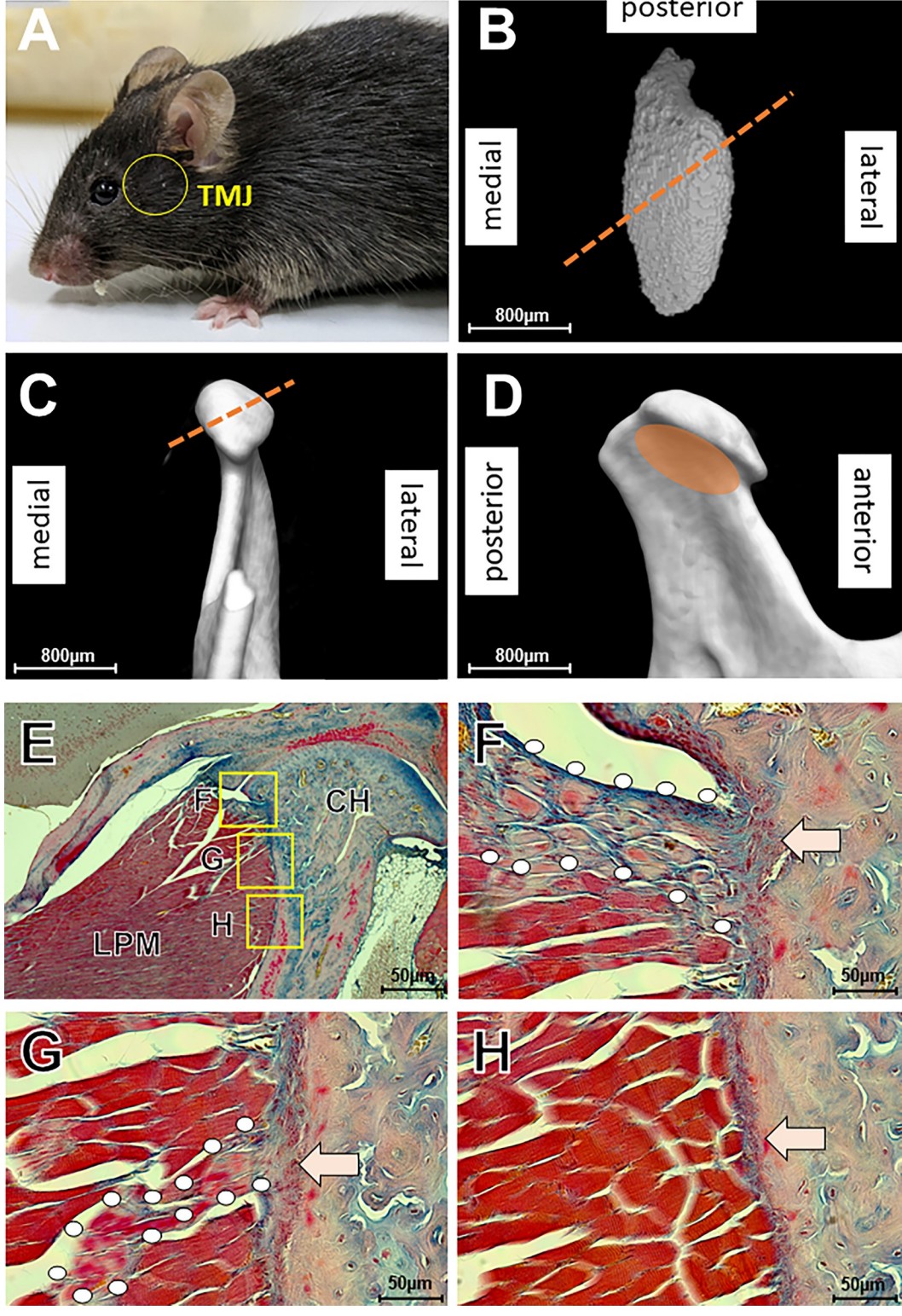

**Fig 1. Mice TMJ enthesis lacks fibrocartilage.** (A) The craniofacial region in mice (yellow circle): TMJ). (B–D) CT images of mice TMJ. The TMJ condyle has a mushroom-like structure. The pterygoid fovea connects to the lateral pterygoid muscle (LPM). (E–H) Azan staining of the TMJ enthesis. F-H are high-magnification views of the squares in E. Dotted lines in B and C show the section plane of the histological images in E–H. Intramuscular tendons (G points) and extramuscular tendons (F points) extend from the pterygoid fovea through fibrous entheses (F, G arrows). The lower part of the LPM is attached to the pterygoid fovea via periosteum without tendons (arrow in H). LPM: lateral pterygoid muscle, CH: condylar head.

and the second was at the LPM's top (Fig 1F). These two tendons were attached to the ptery-goid fovea via "fibrous enthuses" (Fig 1F and 1G). The lower part of the LPM was attached to the pterygoid fovea via the periosteum without tendons (Fig 1H). In mice, thus, LPM featured two types of entheses: "fibrous enthuses" and "periosteal insertions."

## The TMJ enthesis develop in front of the membranous bones and exhibit *Scleraxis during the fetal period*

In E13.5, a condylar anlage positive for ALP$^+$ was positioned posterior-inferior to the LPM anlage marked by desmin$^+$ (Fig 2A, 2D and 2G). Compared to the gap between LPM and the condyle at E13.5 and E14.5, they were much closer at E15.5 (Fig 2A–2F). As the cranial base angle relative to the LPM's long axis progressively decreased from E14 to E18, we demon-strated fetal period LPM rotation (Fig 2D–2J). By E14.5, mesenchymal condensation had formed the condyle (Fig 2K). At E15.5, membranous bones emerged after the TMJ entheses and before the condyle (Fig 2L). Thus, the TMJ enthesis formed anterior to the membranous bones (Fig 2). Jaw movements are clearly observable in fetal mice at E15.5 [23, 24]. Given that a certain LMP maturation level appears closely linked to jaw movement onset, we explored the initiation timing of jaw movements in fetal mice. Jaw movements were observed in 22.2% of mice at E16 and in 100.0% at E18 (Fig 2M and 2N). However, no jaw movement was observed at E14 (Fig 2M). Thus, we chose to scrutinize the TMJ enthesis development in the periods pre- and post-E16. The LMP entheses expressed *Scx* during the fetal period (Fig 2O and 2P), and *Scx* showed a gradual narrowing over time (Fig 2O and 2P). The gap between the muscle and bone in the TMJ gradually narrowed from E14 to E18 (Fig 2Q).

## The expression of SOX9 is negatively regulated during the development of the TMJ enthesis

It is well known that fibrocartilaginous entheses are composed of double-positive $Scx^+$/$Sox9^+$ cells [13]. Furthermore, the $Scx^+$/$Sox9^+$ progenitor cells originate from the enthesis at the angular process of the mandible [25]. Therefore, we hypothesize that the TMJ enthesis would express not only Scx but also *Sox9*. We generated double-transgenic *Sox9CreER$^{T2}$;R26$^{tdTomato/+}$* reporter mice (Fig 3A). Although *tdTomato+* cells were observed at the entheses of mice induced at E13 (Fig 3B and 3C), they were not detected in mice induced at E15 (Fig 3C and 3D). The expression of *Sox9* seems to be negatively regulated during the development of the TMJ enthesis.

To clarify this negative regulation, we performed an immunohistochemical staining of SOX9. Although SOX9 was expressed in the enthesis at E16, this transcription factor's regula-tion decreased by E18 (Fig 3E, 3F and 3J). Runt-related transcription factor 2 (RUNX2) is essential for intramembranous ossification of neural crest-derived cells [26]. Therefore, we hypothesize that the enthesis focusing on intramembranous bone would express RUNX2. Indeed, RUNX2 was expressed in the TMJ entheses at E16, with its expression reduced by E18 (Fig 3G, 3H and 3K). To verify any link between reduced SOX9 and RUNX2 expression and cell proliferation activity, we assessed cell proliferation at entheses (Fig 3I). At E15.5, the den-sity of proliferating cells was 2,382.3 cells/mm$^2$ (Fig 3I and 3L), while at E16.5, it had sharply decreased to 1,258.5 cells/mm$^2$ (Fig 3I and 3L). At E17.5, the density of proliferating cells was 1403.5 cells/mm$^2$ (Fig 3K and 3L). These findings indicated that reduced SOX9 and RUNX2 expression coincided with lower cell proliferation activity in that area.

The muscle-bone attachment site (including enthesis) at E15 showed higher expression lev-els of the Runx2 and Sox9 genes than those at E18 (Fig 3M). In the condyle alone, the relative

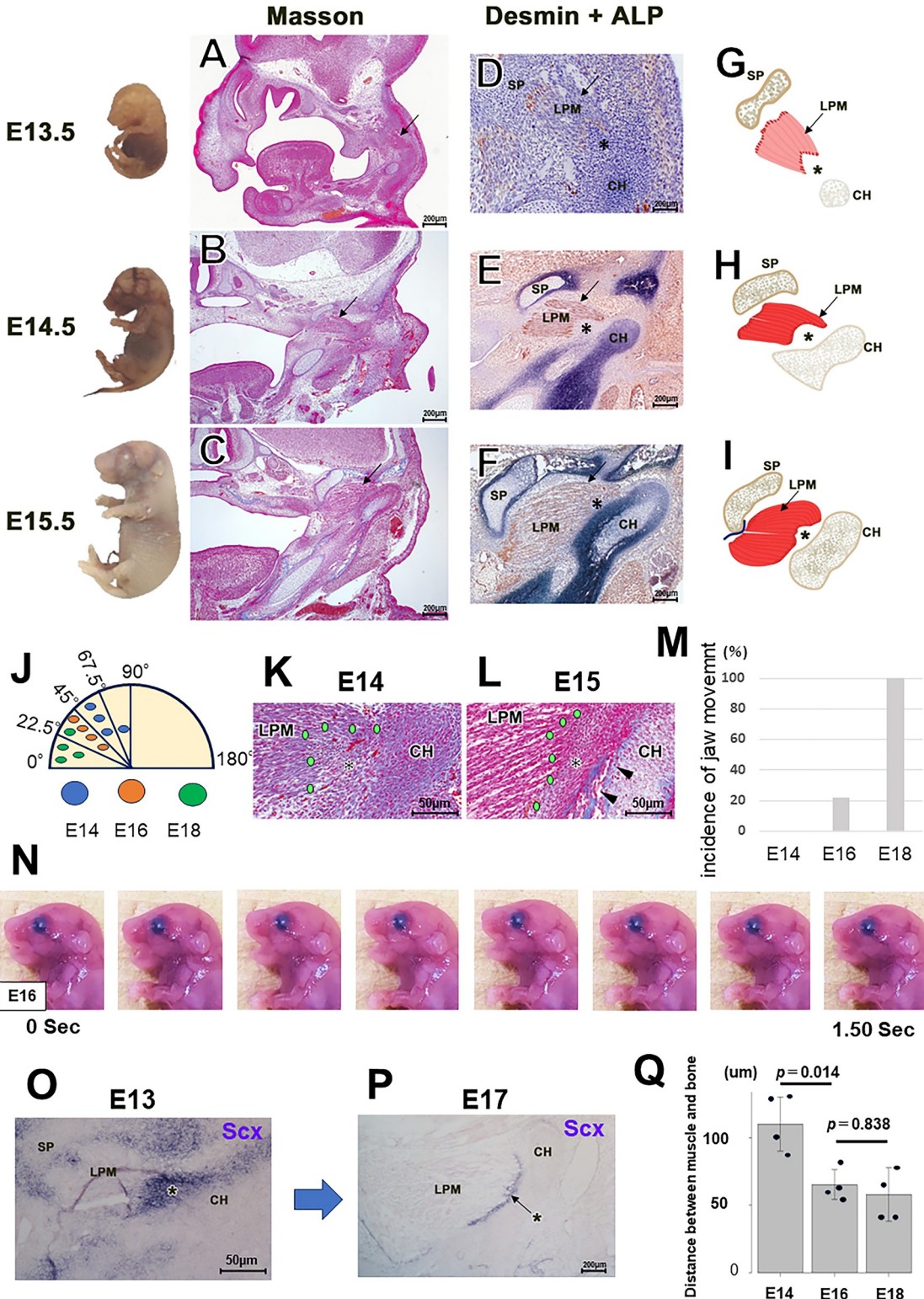

**Fig 2. The TMJ enthesis faces intramembranous bone.** (A–C) Azan staining of the TMJ and surrounding structures. (D–F) Dual-staining development of the TMJ for alkaline phosphatase (ALP) and desmin. (G–I) Schematic representations of the developing TMJ. The black arrows point to the upper margin of the LPM in panels A–I. (J) The skull base angle relative to the LPM's long axis progressively decreases from E14 to E18. (K–L) Development of the TMJ enthesis from E14.4 to E15.5. By E15.5, the membranous bone is seen posterior to the TMJ enthesis (arrow tips), to which the TMJ enthesis attaches. (M–N) Jaw

movements were observed in 22.2% of mice at E16, but in 100.0% at E18 ($n$ = 7 mice per group). (N) Serial photos of jaw movement at E16. (O, P) In situ hybridization of *Scx*. The TMJ enthesis expresses *Scx* in the fetal period. (Q) The gap between muscle and bone in the TMJ narrows progressively from E14 to E16 (n = 4 mice per group; Tukey's multiple comparison test post one-way ANOVA). LPM: lateral pterygoid muscle, CH: condylar head, SP: sphenoid bone, Asterisk: TMJ enthesis.

mRNA level of Runx2 was significantly higher at E14 than at E18. However, the level of Sox9 mRNA was significantly lower at E14 than at E18 (Fig 3N).

## *Sox9* is crucial for enthesis development in the TMJ

β-galactosidase, indicating these cells originated from Wnt*1*-expressing neural crest cells (Fig 4A–4D). To clarify the functional role of Sox9 in TMJ enthesis development, we generated *Wnt1Cre;Sox9$^{flox/+}$* mice (Fig 4E). Fluorescence intensity analysis revealed higher RUNX2 levels in *Wnt1Cre;Sox9$^{flox/+}$* mice compared to controls (Fig 4F–4H). Moreover, RUNX2+ cells were found in the LPM of *Wnt1Cre;Sox9$^{flox/+}$* (Fig 4I–4K), indicating disrupted enthesis formation in the ATM. Therefore, *Sox9* is crucial for early-stage enthesis development.

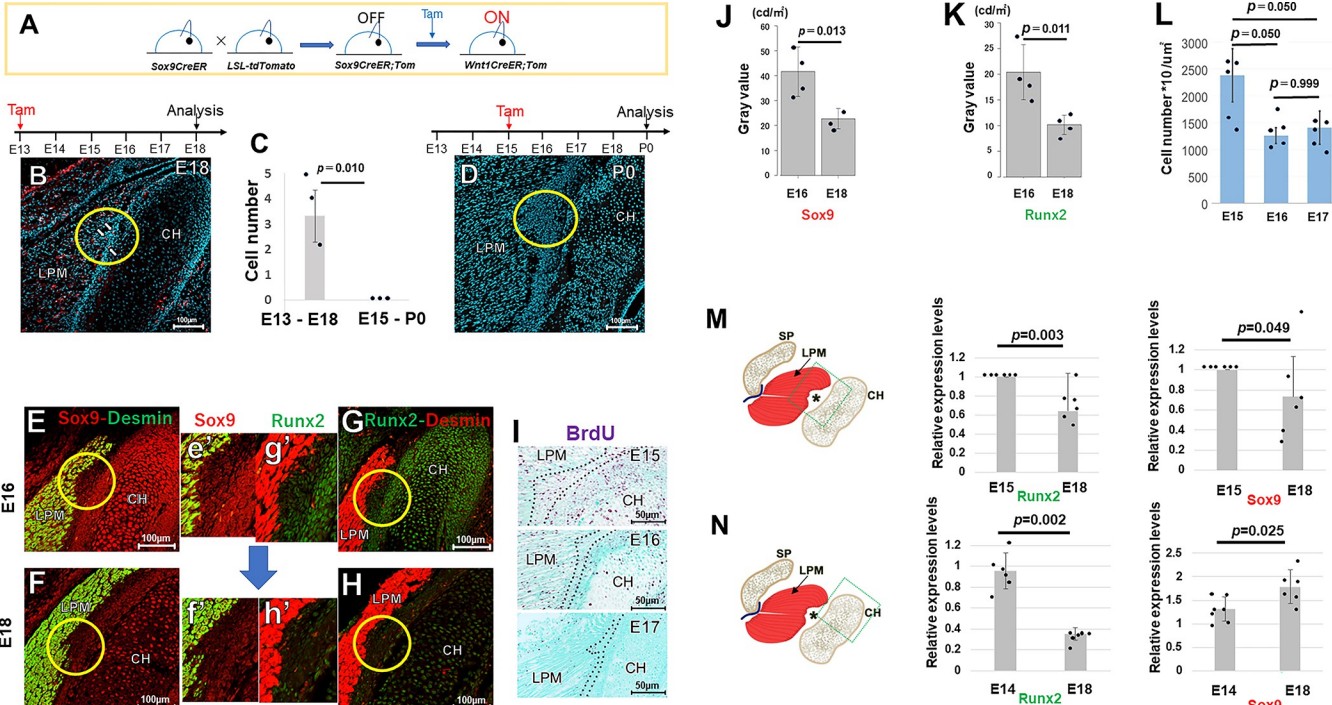

**Fig 3. Transcription factor SOX9 is downregulated over time in the TMJ enthesis.** (A–D) *Sox9CreER$^{T2}$;R26$^{tdTomato/+}$* reporter mice. We detected *tdTomato+* cells (white arrows) at the enthesis of mice induced at E13, but not any tdTomato+ cells in mice induced at E15 ($n$ = 3 mice per group; unpaired Student's t-test). (E–J) Immunohistochemical staining for SOX9 and RUNX2. While SOX9 is expressed in the enthesis at E16, this transcription factor is markedly reduced by E18 ($n$ = 3 mice per group, Tukey's multiple comparison test post one-way repeated measures ANOVA). RUNX2 is expressed in TMJ entheses at E16, with a significant reduction by E18 ($n$ = 3 mice per group, Tukey's multiple comparison test post one-way repeated measures ANOVA). (K–L) BrdU staining for identifying proliferating cells in the TMJ enthesis. Cells with proliferative activity at E16 and E17 are significantly decreased compared to E15 ($n$ = 5 mice per group, Tukey's multiple comparison test post one-way ANOVA; repeated measures). (M) mRNA expression at the muscle-bone attachment site (including enthesis). This site at E15 shows higher levels of expression of the Runx2 and Sox9 genes than those at E18 ($n$ = 6 mice per group, unpaired Student's t-test). (N) mRNA expression in the condyle alone. Runx2 expression is significantly higher at E14 than at E18. The level of Sox9 expression is significantly lower at E14 than at E18 ($n$ = 6 mice per group, unpaired Student's t-test). LPM: lateral pterygoid muscle, CH: condylar head. SP: sphenoid.

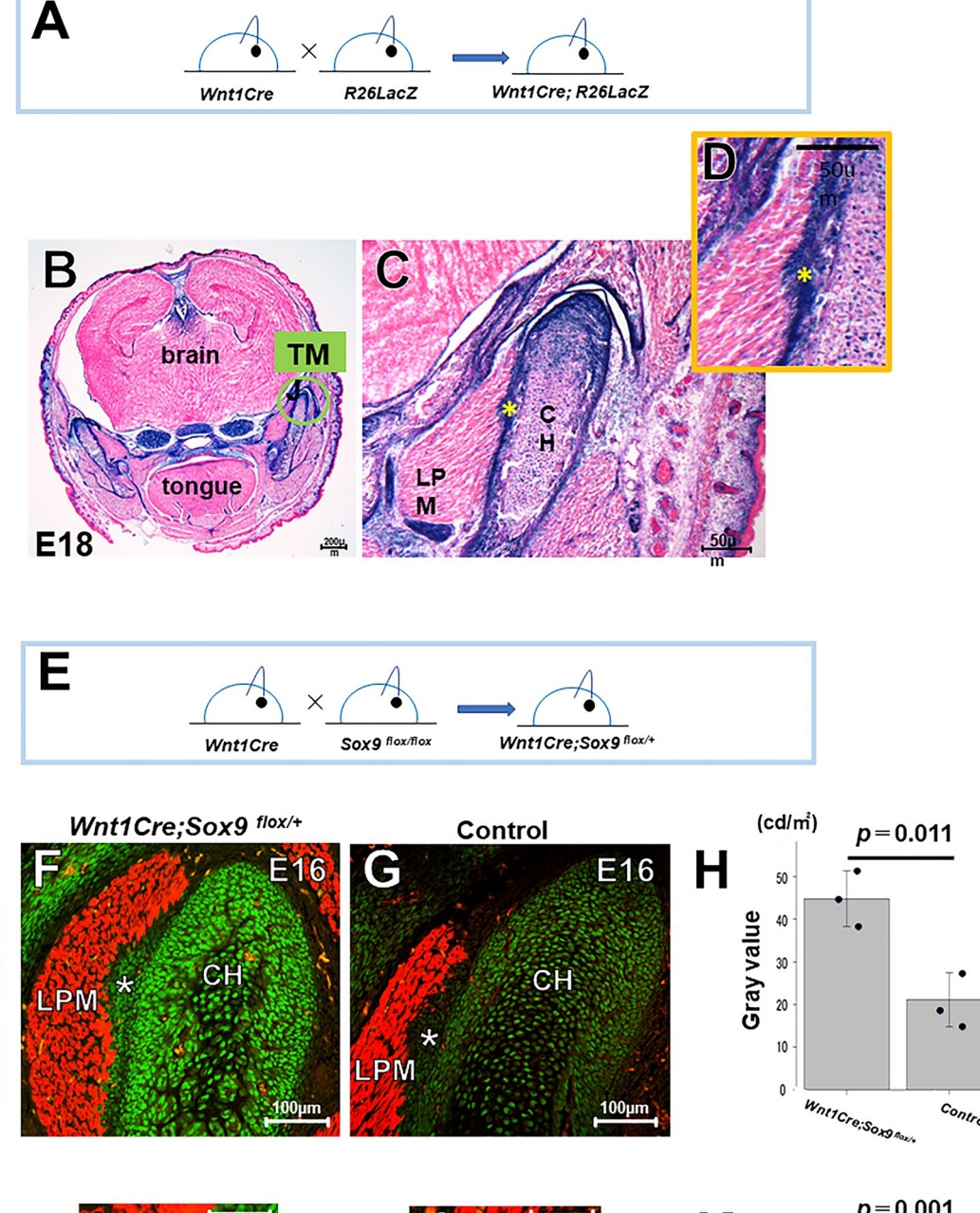

**Fig 4. *Wnt1Cre;*$^{Sox9flox/+}$ mice exhibit deformation of the TMJ entheses.** (A–D) *Wnt1Cre;R26*$^{LacZ/+}$ mice. The TMJ enthesis expresses β-galactosidase. (E–K) *Wnt1Cre;Sox9*$^{flox/+}$ mice. Fluorescence intensity analysis reveals that RUNX2 expression is significantly higher in *Wnt1Cre;Sox9*$^{flox/+}$ mice compared to control mice (F-H) *(n* = with 3 mice per group, unpaired Student's t-test). Some RUNX2$^+$ cells are found in the LPM of *Wnt1Cre;Sox9*$^{flox/+}$ mice (I-K)*(n* = with 3 mice per group, unpaired Student's t-test). LPM: lateral pterygoid muscle, CH: condylar head, Asterisk: TMJ enthesis.

### There is a diminished SOX9 expression area in the enthesis in contact with intramembranous bones, as seen in the TMJ entheses

To clarify the typical development of the TMJ enthesis, we studied the entheses in two upper limbs(①Shoulder: Attachment of the supraspinatus muscle to the humerus and attachment of the deltoid muscle to the clavicle; ②Elbow: Attachment of the triceps brachii muscle to the ulna (Fig 5).

On the shoulder (①) at E16.5, a SOX9+ enthesis was identified between the supraspinatus muscle and the humerus in the COL2+ cartilage cell surface layer (Fig 5A(a)–5A(d)), yet in the same section, there was scarcely any SOX9 expression in the enthesis between the deltoid muscle and the clavicle (Fig 5A(a)–5A(d)). The luminance of SOX9+ cells in the same section was below 100 at the enthesis between the supraspinatus muscle and the humerus, yet above 100 at one of the entheses between the deltoid muscle and the clavicle (Fig 5A(e)), indicating high Sox9 expression at the supraspinatus enthesis and low at the deltoid entheses. At E13.5, however, Sox9 was expressed in the deltoid muscle enthesis (Fig 5A(f)–5A(i)).

At the elbow (②) in E15.5, two types of SOX9+ cells were apparent at the enthesis: flat cells with polarity, akin to tendon tissues, and elliptical cells (Fig 5B(a)–5B(d)). At E16.5, the medial head of the brachial triceps muscle had formed a covering over the olecranon cartilage (Fig 5B(e), arrow), and newly emerged elliptical SOX9+ cells were noted at the enthesis (Fig 5B(f), 5B(h) and 5B(a')), while the tendon tissue attachment region to the cartilage consisted of flat Sox9+ cells with polarity, akin to tendinous tissues (Fig 5B(h) and 5B(b')).

We also observed a diminished SOX9 expression area at the enthesis in contact with the clavicle, similar to the entheses of the TMJ.

## Discussion

This study showed that the development of fibrous entheses and periosteal insertions is unique and markedly distinct from the previously reported fibrocartilage entheses: (1) fibrous entheses and periosteal insertions were adjacent to intramembranous bone instead of cartilage and expressed Scx. (2) SOX9 expression diminished during the development of these entheses.

The mandibular condyle consists of the head, the articular surface, and the neck (the narrow part that supports it). The front part of the neck has a depression known as the pterygoid fossa, which acts as an attachment point for the LPM tendon [27]. While the development of entheses in the head is not well understood, previous studies have shown that muscle, tendon, and bone formation in the head differs from that in the limbs and trunk [28, 29], indicating a potentially unique mechanism for enthesis formation in the head. The present study showed that the LPM rotated during the fetal period. At E13.5, the condylar anlage was located below the LPM, and then rapidly extended in the period from E14.5 to E15.5. The condylar cartilage is a secondary cartilage that undergoes rapid hypertrophy accompanying the development of the condyle [29–31]. This condyle elongation associated with the cartilage hypertrophy seems to facilitate LPM rotation. Because the tendon anlage with the LPM also rotates, formation of the TMJ entheses is complex.

While Sox9 expression in entheses is well known, Runx2 expression in entheses remains unclear. However, some researchers have reported Runx2 expression in entheses. A study using an Achilles enthesis organ culture model indicated that mechanical stress increased Runx2 expression in the enthesis [32]. Kuntz et al. [33] provided a list of genes whose expression was enriched in enthesis and identified candidate transcription factors included Runx2 and Sox9. Komori et al. [34] reported that Runx2 is essential for osteoblast differentiation and chondrocyte maturation. In the present study, we identified Runx2 expression in the TMJ enthesis. However, this transcription factor decreased shortly before birth. Since Runx2

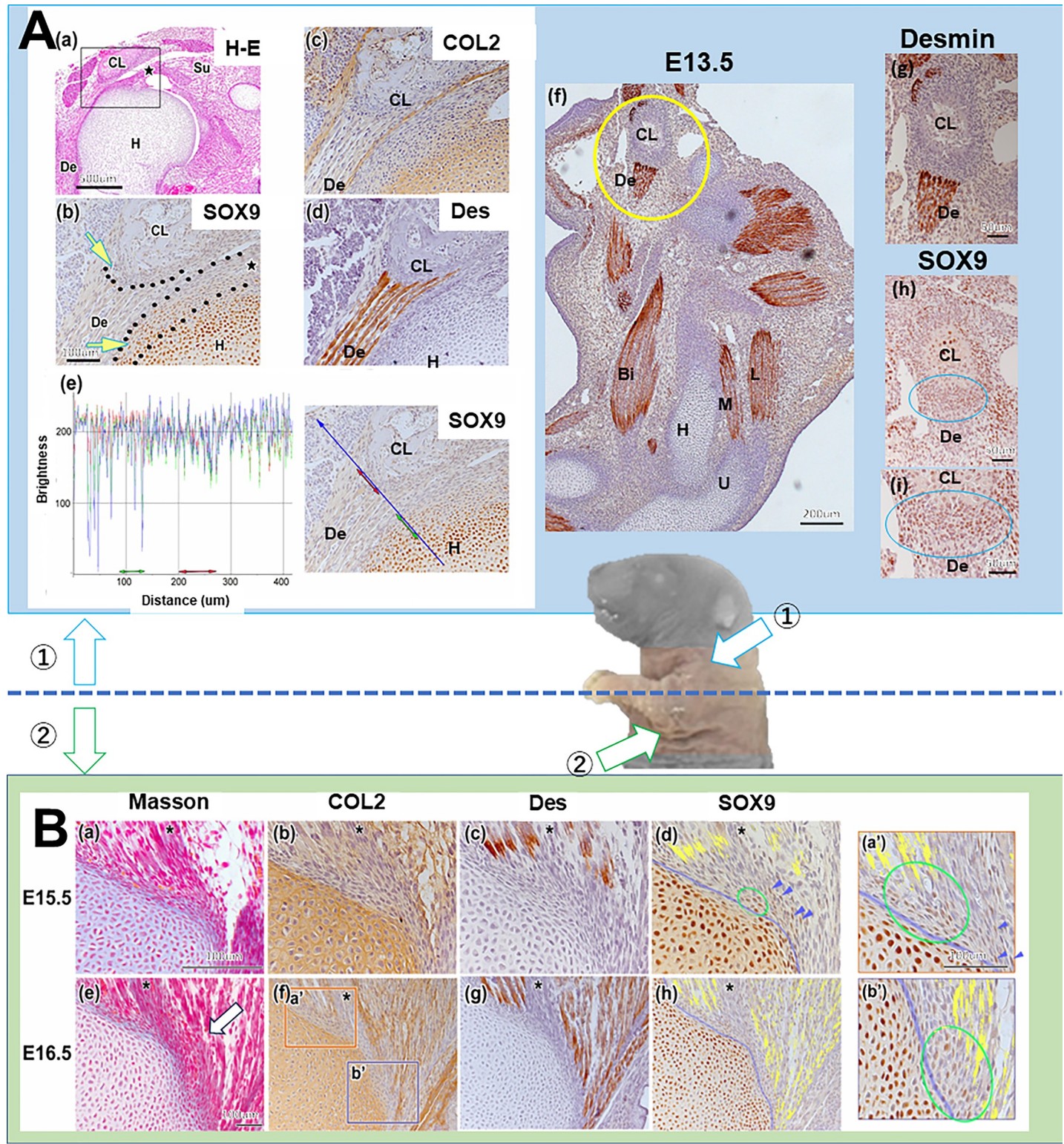

**Fig 5. SOX9 downregulation were identified in the enthesis in contact with the clavicle.** (A) Shoulder: Attachment of the supraspinatus muscle to the humerus and attachment of the deltoid muscle to the clavicle. (b)–(d) represent the same magnification. A SOX9+ enthesis is apparent between the supraspinatus muscle and the humerus, whereas no SOX9 expression is found in the enthesis between the deltoid muscle and the clavicle ((a)–(e)). At E13.5, Sox9 is expressed in the deltoid muscle enthesis ((f)–(i)). (B) Elbow: Attachment of the brachial triceps muscle to the ulna. (a)–(d) ((e)–(h)) are at the same magnification. (a') is the same magnification as (b'). In E15.5 and 16.5, Sox9+ is observed in the elbow enthesis ((a)–(h) and (a')–(b')). At E15.5, SOX9+ cell morphology in the tendon attachment region presents as two

types: flattened (blue arrowheads), showing polarity like tendon tissues, and elliptical (ovals) ((d)). At E16.5, elliptically shaped Sox9+ cells can be found between cartilage and muscle ((a')–(b'); ovals). The tissue attachment region to cartilage consists of flat Sox9+ cells with a polarity akin to tendon cells ((**a'**); blue arrowheads). Bi: biceps brachii muscle, CL: clavicle, De: Deltoid muscle, H: Huemuls, M: medial head of the triceps brachii muscle, L: lateral head of the triceps brachii muscle, asterisk: tendon of the triceps brachii muscle. U: ulna.

expression was elevated as a result of Sox9 knockout, it appears to be crucial for early TMJ enthesis development. This study is the first to have demonstrated Runx2 expression in a developing enthesis.

Our study showed that Sox9 and Runx2 expression decreased over time in TMJ enthesis development. However, Runx2 expression did not decrease in the TMJ entheses of *Wnt1Cre; Sox9^flox/+^*. Since *Sox9* knockout hinders TMJ enthesis development, enthesis maintained RUNX2 expression until E16. The negative double regulation of RUNX2 and SOX9 might facilitate the maturation of entheses. Proliferative cells were found at their entheses on E15, indicating that RUNX2 and SOX9 expression in TMJ entheses is crucial for cell division. Our previous study indicates that TMJ entheses distinctly express SOX9 at E13 [19].

A notable finding was the reduced SOX9 expression in the TMJ enthesis area and a consequent decline in cell proliferation activity there from E16.5. This implied a potential link between SOX9 expression and cell proliferation. Moreover, SOX9 expression levels at E16.5 were high in the supraspinatus attachment region to the scapula, yet low in the deltoid attachment area to the clavicle. As established in prior research, an enthesis is a tendon's attachment to bone, classified into fibrous and fibrocartilaginous types [35]. A fibrous enthesis directly connects fibrous tissue to bone and periosteum, typically at the shafts of long bones. A fibrocartilaginous enthesis is an attachment to bone via a fibrocartilage layer, found in the epiphysis [5, 36]. The epiphysis forms via endochondral ossification, while the diaphysis's bone collar arises from membranous ossification [37]. In this study, we show that the TMJ enthesis and the deltoid enthesis are membranous bones and that SOX9 expression decreased in the TMJ enthesis during the fetal period. These findings suggest that the temporal pattern of SOX9 expression varies with the ossification development mode, whether it is linked to fibrous or fibrocartilaginous enthesis. Since the entheses of the TMJ are fibrous or periosteal insertions (Fig 1), SOX9 seems to have been downregulated during fetal development. Ideo et al. [38] showed that Sox9 is expressed in the mouse fibrocartilaginous entheses 3 weeks postnatally.

Chen et al. investigated PTH-related protein (PTHrP) expression in young and adult animals using PTHrP-lacZ knockin mice, and demonstrated that fibrous entheses and periosteal insertions express PTHrP [39]. This group also demonstrated that mechanical stress is a key for maintaining the homeostasis of fibrous enhtheses, because the levels of PTHrP expression are reduced in the fibrous entheses of tail-suspended mice [40]. Wang et al. showed that PTHrP regulates periosteal/intramembranous bone cell activity in fibrous entheses. However, little is known about the development of fibrous entheses and periosteal insertions [41]. The present study has demonstrated that the development of fibrous entheses and periosteal insertions differs from that of fibrocartilaginous entheses. Although fibrous entheses are not clinically important, further study is needed.

## Supporting information

**S1 Table. Primary antibodies for immnofluorescence.**
(XLSX)

**S2 Table. Secondary antibodies for immnofluorescence.**
(XLSX)

**S3 Table. Primer assay ID and amplicon length of RT-PCR.**
(DOCX)

## Acknowledgments

We sincerely thank all the staff of the Department of Anatomy and the Cornea Center at Tokyo Dental College, for their wholehearted cooperation. We are grateful to Keko Yokoyama and Hideyuki Matsuzawa, the Medical Science College Office, Tokai University.

## Author Contributions

**Conceptualization:** Masahito Yamamoto.

**Data curation:** Asahi Kitamura, Masahito Yamamoto, Hidetomo Hirouchi.

**Formal analysis:** Asahi Kitamura, Masahito Yamamoto, Genji Watanabe, Shuichiro Taniguchi, Sayo Sekiya, Satoshi Ishizuka.

**Funding acquisition:** Masahito Yamamoto, Shinichi Abe.

**Investigation:** Asahi Kitamura, Masahito Yamamoto, Genji Watanabe, Shuichiro Taniguchi, Sayo Sekiya.

**Methodology:** Juhee Jeong, Kazunari Higa.

**Project administration:** Shuichiro Yamashita, Shinichi Abe.

**Resources:** Shinichi Abe.

**Software:** Satoshi Ishizuka.

**Writing – original draft:** Masahito Yamamoto.

**Writing – review & editing:** Shinichi Abe.

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
