## [Decision Letter · Decision Letter 0]

6 Dec 2023

PONE-D-23-38678Downregulation of SOX9 expression in developing entheses adjacent to intramembranous bonePLOS ONE

Dear Dr. Yamamoto,

Thank you for submitting your manuscript to PLOS ONE. After careful consideration, we feel that it has merit but does not fully meet PLOS ONE’s publication criteria as it currently stands. Therefore, we invite you to submit a revised version of the manuscript that addresses the points raised during the review process.

We look forward to receiving your revised manuscript.

Kind regards,

Gianpaolo Papaccio, M.D., Ph.D.

Academic Editor

PLOS ONE

Journal Requirements:

2. To comply with PLOS ONE submissions requirements, in your Methods section, please provide additional information regarding the experiments involving animals and ensure you have included details on (1) methods of anesthesia and/or analgesia, and (2) efforts to alleviate suffering.

   "This study was supported by a grant for Assistance in Joint Research (20K10191: Masahito Yamamoto and 20K09895: Shinichi Abe) with the Community Program in Life Sciences from the Ministry of Education, Culture, Sports, Science and Technology of Japan."

Additional Editor Comments:

This manuscript has been found to be of interest, pending some points to be addressed.

The Authors must add some experiments in order to ameliorate it, as suggested by the two reviewers.

They must take into account all the points and suggestions and, in the rebuttal letter along with the revised manuscript, they must highlight all the amendments they have done.

Reviewers' comments:

Reviewer's Responses to Questions

**Comments to the Author**

1. Is the manuscript technically sound, and do the data support the conclusions?

Reviewer #1: Yes

Reviewer #2: Yes

2. Has the statistical analysis been performed appropriately and rigorously? 

Reviewer #1: Yes

Reviewer #2: Yes

3. Have the authors made all data underlying the findings in their manuscript fully available?

Reviewer #1: Yes

Reviewer #2: Yes

4. Is the manuscript presented in an intelligible fashion and written in standard English?

Reviewer #1: Yes

Reviewer #2: No

5. Review Comments to the Author

Reviewer #1: In this paper Authors showed that SOX9 expression is negatively regulated during TMJ enthesis development, unlike fibrocartilage entheses which are modularly formed by SCX and SOX9 positive progenitors.

The paper is interesting, the experiments are well conducted and the data are well presented.

In any case, to reinforce their hypotheses and add importance to the study, Authors must add some molecular data, performing qPCR and WB experiments.

Moreover, Author must improve discussion section.

Reviewer #2: The Authors aimed to evaluate the role of SOX9 in the TMJ enthesis development using murine models and demonstrated that Sox9 and Runx2 expression decreased over time in TMJ enthesis development except for fibrocartilage entheses where there are progenitors positive to Sox9.

The results are good and clear as well as the figures. Although this, Introduction and discussion must be completely revised. In the Introduction, it is important to indicate the purpose of the study. The Discussion must be more organic. The authors must delete the reference to figure 1 in this section.

Moreover, in my opinion, results obtained need to be validated with gene expression data.

6. PLOS authors have the option to publish the peer review history of their article (what does this mean?). If published, this will include your full peer review and any attached files.

Reviewer #1: No

Reviewer #2: No

---

## [Author Response · Author response to Decision Letter 0]

23 Feb 2024

Reviewer #1

Thank you for your positive evaluation of our work. According to your comment, we have revised our manuscript.

- In any case, to reinforce their hypotheses and add importance to the study, the Authors must add some molecular data, performing qPCR and WB experiments.

Although it is difficult for us to extract TMJ entheses alone, we performed qPCR. New panels were added in Figure 3.

- Moreover, the Author must improve the discussion section.

In the discussion section, we added sentences about Runx2 and PTHrP expressions in entheses as follows.

P11 L280 

The mandibular condyle consists of the head, the articular surface, and the neck (the narrow part that supports it). The front part of the neck has a depression known as the pterygoid fossa, which acts as an attachment point for the LPM tendon [27]. While the development of entheses in the head is not well understood, previous studies have shown that muscle, tendon, and bone formation in the head differs from that in the limbs and trunk [28, 29], indicating a potentially unique mechanism for enthesis formation in the head. The present study showed that the LPM rotated during the fetal period. At E13.5, the condylar anlage was located below the LPM, and then rapidly extended in the period from E14.5 to E15.5. The condylar cartilage is a secondary cartilage that undergoes rapid hypertrophy accompanying the development of the condyle [29-31]. This condyle elongation associated with the cartilage hypertrophy seems to facilitate LPM rotation. Because the tendon anlage with the LPM also rotates, formation of the TMJ entheses is complex. 

While Sox9 expression in entheses is well known, Runx2 expression in entheses remains unclear. However, some researchers have reported Runx2 expression in entheses. A study using an Achilles enthesis organ culture model indicated that mechanical stress increased Runx2 expression in the enthesis [32]. Kuntz et al. [33] provided a list of genes whose expression was enriched in enthesis and identified candidate transcription factors included Runx2 and Sox9. Komori et al. [34] reported that Runx2 is essential for osteoblast differentiation and chondrocyte maturation. In the present study, we identified Runx2 expression in the TMJ enthesis. However, this transcription factor decreased shortly before birth. Since Runx2 expression was elevated as a result of Sox9 knockout, it appears to be crucial for early TMJ enthesis development. This study is the first to have demonstrated Runx2 expression in a developing enthesis.

P13　L325

Chen et al. investigated PTH-related protein (PTHrP) expression in young and adult animals using PTHrP-lacZ knockin mice, and demonstrated that fibrous entheses and periosteal insertions express PTHrP [40]. This group also demonstrated that mechanical stress is a key for maintaining the homeostasis of fibrous enhtheses, because the levels of PTHrP expression are reduced in the fibrous entheses of tail-suspended mice [41]. Wang et al. showed that PTHrP regulates periosteal/intramembranous bone cell activity in fibrous entheses. However, little is known about the development of fibrous entheses and periosteal insertions[42]. The present study has demonstrated that the development of fibrous entheses and periosteal insertions differs from that of fibrocartilaginous entheses. Although fibrous entheses are not clinically important, further study is needed.

Reviewer #2

Thank you for your positive evaluation of our work. According to your comment, we have revised our manuscript.

- Introduction and discussion must be completely revised. In the Introduction, it is important to indicate the purpose of the study. The Discussion must be more organic. 

We added several sentences and the purpose of this study in the introduction section. 

P2 L51

Because no specific marker of tendon development has yet been identified, it has long been unclear how tendon-bone interfaces develop. In the early 2000s, several groups demonstrated that scleraxis (SCX), a bHLH transcription factor, is expressed throughout tendon differentiation [8,9]. To determine the function of Scx, Murchison et al. [10] generated Scx–/– mice and found that they exhibited severe tendon defects. This led to further investigations of the role of Scx in development of the tendon-bone interface. Blitz et al. [11] showed that bone morphogenetic protein 4 (BMP4) acts downstream of Scx, and that deletion of Bmp4 in tendons has a marked effect on enthesis development. In 2013, two research groups demonstrated that progenitors double-positive for Scx and Sox9 (a regulator of cartilage formation) form entheses under the regulation of TGFβ and BMP4 signaling [12,13]. In mice, Schwartz et al. [14] revealed a population of Hedgehog-responsive cells in the developing enthesis that is distinct from tendon and chondrocytes. Although previous studies have clarified the development of fibrocartilaginous entheses, little is known about that in “fibrous entheses” and “periosteal insertions”.

The temporomandibular joint (TMJ) is a hinge-type synovial joint that connects the mandible (lower jaw) to the temporal bone. Specifically, it forms an articulation between the condylar process (mandible) and the mandibular fossa (temporal bone). The articular disc is a thin oval plate composed of non-vascular fibrous connective tissue located between the two [15,16]. The lateral pterygoid muscle, one of the four masticatory muscles, attaches to the anterior part of the condylar process and the articular disc [17,18]. Entheses in the human TMJ are unique because they comprise all types of entheses [19]. However, it is unclear how they develop in the TMJ. This study aimed to clarify the development of “fibrous entheses” and “periosteal insertions” by investigating the TMJ.

The discussion section was revised.

P11 L280 

The mandibular condyle consists of the head, the articular surface, and the neck (the narrow part that supports it). The front part of the neck has a depression known as the pterygoid fossa, which acts as an attachment point for the LPM tendon [27]. While the development of entheses in the head is not well understood, previous studies have shown that muscle, tendon, and bone formation in the head differs from that in the limbs and trunk [28, 29], indicating a potentially unique mechanism for enthesis formation in the head. The present study showed that the LPM rotated during the fetal period. At E13.5, the condylar anlage was located below the LPM, and then rapidly extended in the period from E14.5 to E15.5. The condylar cartilage is a secondary cartilage that undergoes rapid hypertrophy accompanying the development of the condyle [29-31]. This condyle elongation associated with the cartilage hypertrophy seems to facilitate LPM rotation. Because the tendon anlage with the LPM also rotates, formation of the TMJ entheses is complex. 

While Sox9 expression in entheses is well known, Runx2 expression in entheses remains unclear. However, some researchers have reported Runx2 expression in entheses. A study using an Achilles enthesis organ culture model indicated that mechanical stress increased Runx2 expression in the enthesis [32]. Kuntz et al. [33] provided a list of genes whose expression was enriched in enthesis and identified candidate transcription factors included Runx2 and Sox9. Komori et al. [34] reported that Runx2 is essential for osteoblast differentiation and chondrocyte maturation. In the present study, we identified Runx2 expression in the TMJ enthesis. However, this transcription factor decreased shortly before birth. Since Runx2 expression was elevated as a result of Sox9 knockout, it appears to be crucial for early TMJ enthesis development. This study is the first to have demonstrated Runx2 expression in a developing enthesis.

P13　L325

Chen et al. investigated PTH-related protein (PTHrP) expression in young and adult animals using PTHrP-lacZ knockin mice, and demonstrated that fibrous entheses and periosteal insertions express PTHrP [40]. This group also demonstrated that mechanical stress is a key for maintaining the homeostasis of fibrous enhtheses, because the levels of PTHrP expression are reduced in the fibrous entheses of tail-suspended mice [41]. Wang et al. showed that PTHrP regulates periosteal/intramembranous bone cell activity in fibrous entheses. However, little is known about the development of fibrous entheses and periosteal insertions [42]. The present study has demonstrated that the development of fibrous entheses and periosteal insertions differs from that of fibrocartilaginous entheses. Although fibrous entheses are not clinically important, further study is needed.

- The authors must delete the reference to Figure 1 in this section.

We deleted the references.

- Moreover, in my opinion, the results obtained need to be validated with gene expression data.

Although it is difficult for us to extract TMJ entheses alone, we performed qPCR. New panels were added in Figure 3.

---

## [Decision Letter · Decision Letter 1]

11 Mar 2024

Downregulation of SOX9 expression in developing entheses adjacent to intramembranous bone

PONE-D-23-38678R1

Dear Dr. Yamamoto,

We’re pleased to inform you that your manuscript has been judged scientifically suitable for publication and will be formally accepted for publication once it meets all outstanding technical requirements.

Kind regards,

Gianpaolo Papaccio, M.D., Ph.D.

Academic Editor

PLOS ONE

Additional Editor Comments (optional):

Reviewers' comments:

Reviewer's Responses to Questions

**Comments to the Author**

1. If the authors have adequately addressed your comments raised in a previous round of review and you feel that this manuscript is now acceptable for publication, you may indicate that here to bypass the “Comments to the Author” section, enter your conflict of interest statement in the “Confidential to Editor” section, and submit your "Accept" recommendation.

Reviewer #1: All comments have been addressed

Reviewer #2: All comments have been addressed

2. Is the manuscript technically sound, and do the data support the conclusions?

Reviewer #1: (No Response)

Reviewer #2: (No Response)

3. Has the statistical analysis been performed appropriately and rigorously? 

Reviewer #1: (No Response)

Reviewer #2: (No Response)

4. Have the authors made all data underlying the findings in their manuscript fully available?

Reviewer #1: (No Response)

Reviewer #2: (No Response)

5. Is the manuscript presented in an intelligible fashion and written in standard English?

Reviewer #1: (No Response)

Reviewer #2: (No Response)

6. Review Comments to the Author

Reviewer #1: (No Response)

Reviewer #2: The authors have adequately addressed my comments, point by point , raised in a previous round of review.

7. PLOS authors have the option to publish the peer review history of their article (what does this mean?). If published, this will include your full peer review and any attached files.

Reviewer #1: No

Reviewer #2: No

---

## [Editor Report · Acceptance letter]

18 Mar 2024

PONE-D-23-38678R1 

PLOS ONE

Dear Dr. Yamamoto, 

I'm pleased to inform you that your manuscript has been deemed suitable for publication in PLOS ONE. Congratulations! Your manuscript is now being handed over to our production team.

Kind regards, 

on behalf of

Prof. Gianpaolo Papaccio 

Academic Editor

PLOS ONE